# Lignin Electrolysis at Room Temperature on Nickel Foam for Hydrogen Generation: Performance Evaluation and Effect of Flow Rate

Mohmmad Khalid, Biswajit Samir De, Aditya Singh and Samaneh Shahgaldi *

Institut d'Innovations en Écomatériaux, Écoproduits et Écoénergies, Université du Québec à Trois-Rivières, 3351 Boul. des Forges, Trois-Rivières, QC G8Z 4M3, Canada
* Correspondence: samaneh.shahgaldi@uqtr.ca

**Abstract:** Water electrolysis is a thermodynamically energy-intensive process. One approach employed to make water electrolysis kinetically favorable is replacing the oxygen evolution reaction (OER) at the anode by facile electrooxidation of biomass-feedstocks such as ethanol, methanol, glycerol, and lignin due to the presence of readily oxidizable functional groups. In this work, we report a simplistic approach for hydrogen generation by lignin electrolysis, utilizing a low-cost nickel foam as both anode and cathode sandwiched with hydroxide ion ($OH^-$) exchange membrane in a 3D printed reactor. The performance of the lignin electrolysis was analyzed under various flow rates of anolyte (lignin)/catholyte (KOH) in the anode and cathode chambers. The lignin electrolysis outcompetes traditional water electrolysis by achieving higher current density in the applied voltage range from 0 to 2.5 V at room temperature. The charge transfer resistance for the lignin electrolysis is lower than that of the water electrolysis characterized by impedance spectroscopy. The enhanced current density from the lignin electrolysis at low overvoltage has been presumed from the oxidation of reactive functional groups present in the lignin, facilitating faster electron transfer. Moreover, the hydrogen production rate calculated from the chronoamperometry test of the lignin electrolysis is 2.7 times higher than that of water electrolysis. Thus, the electrochemical oxidation of lignin can potentially lower the capital cost of renewable hydrogen production.

**Keywords:** biomass; lignin; electrolysis; hydrogen production

## 1. Introduction

The continuous rise of global demand for energy and the diminution of fossil fuel reserves are driving the scientific community to search for alternative energy diversification options that are environmentally friendly and sustainable. To this end, hydrogen—a zero-emission fuel, is touted as an alternative to fossil fuels to meet our energy demand [1]. Hydrogen production through water electrolysis is a promising way to overcome environmental issues. Water electrolysis is typically performed under an acidic or alkaline medium with a large voltage (1.8–2.5 V) at elevated temperature. Each medium, however, has its limitations; for example, proton exchange membrane (PEM) based electrolyzers require scarce and expensive noble metal catalysts such as Pt, Ir, and Ru [2,3]. Despite using non-precious metal catalysts, an alkaline electrolyzer needs auxiliary components that intrinsically limit the system's lifetime under concentrated KOH solution (20–40%) and shows relatively low current density (from 0.2–0.4 A $cm^{-2}$) and low discharge hydrogen pressure (1–30 bar). Another approach for water electrolysis has been demonstrated recently using an anion exchange membrane (AEM), which works similarly to PEM but with a flow of $OH^-$ from the cathode to the anode during operation [4]. Nevertheless, to achieve the best performance, both PEM and AEM water electrolyzers require bipolar plates, electrocatalysts, porous transport layers (PTL), and current collectors, which further accounts for the high cost of hydrogen produced by water splitting [5,6]. These limiting factors for water splitting in

conventional devices require the low-cost of their key components, fast reaction kinetics, and no corrosion issues. In addition, the oxygen evolution reaction (OER) that takes place at the anode contributes mainly to the overvoltage compared to the hydrogen evolution reaction (HER) at the cathode [7–9].

To this end, developing bio-derived hydrogen by replacing water oxidation in the anode with biomass-derived organic feedstocks (ethanol, methanol, glycerol, lignin, etc.) is a desirable alternative [10–14]. The organic molecules undergo electrooxidation at lower overvoltage than the water oxidation because of readily oxidizable functional groups, which results in faster electron transfer, thereby enhancing the thermodynamics and kinetics of water electrolysis [15]. Lignin is a biopolymer produced as waste in the form of black liquor in the pulping industry. Therefore, the valorization of lignin in value-added aromatic products and hydrogen generation is regarded as the most attractive way to use lignin waste as a resource, which are currently obtained from fossil fuels [16,17].

In the past few years, several achievements have been made in the electrooxidation of lignin for hydrogen production. Overwhelming efforts have been conducted so far with lignin electrooxidation based on classical three-electrode cell systems, which are limited to pre-assessment of the catalyst and are not typically representative of the practical application due to complicated mass transport and reaction conditions [18–25]. Caravaca et al. [16] electrolyzed lignin in an electrolyzer with a hydroxyl ion conductor membrane (Fumapem) in the middle of the Pt–Ru anode and Pt/C cathode under direct flow of lignin electrolyte. They achieved a lower overvoltage of ~0.45 V than thermodynamically favored water electrolysis (~1.23 V) at elevated temperature (80 °C) [16]. The same group later examined the effect of nickel-based catalyst (Ni/C) on electrooxidation of lignin and its model molecule 2-phenoxyethanole and found that the $Ni(OH)_2$ phase and NiOOH species were responsible for the electrochemical oxidation of primary -OH linked to aromatic functional groups and terminal -OH bonds linked to β-O-4 species, respectively [17]. These contributions were significant in developing lignin electrolysis technology [16,17,26]. However, lignin electrolysis on expensive noble metals and their complex synthesis process while attaining lower current density at elevated temperatures with commercial bulky electrolyzer hardware is undesirable. The key bottleneck in biomass electrolysis technology is to develop a suitable flow reactor using robust 3D-structured anodes and cathodes made of non-precious metals that are economically competitive and more efficient than conventional water electrolysis systems while achieving high currents at low temperatures with long-term operation. The reactor design for lignin electrolysis should be scalable and modular to integrate it with downstream processing for precious byproducts recovery by implementing process intensification strategies [27–31].

This study is an extension of our previous work on lignin electrooxidation for hydrogen generation, where trimetallic nanoalloy of NiFeCo incorporated in phosphidated nitrogen-doped carbon was successfully implemented and showed an enhanced rate of hydrogen production in the presence of lignin compared to conventional water electrolysis in alkaline media [32]. In the present work, we fabricated a reactor compatible with lignin electrolysis using 3D printing. The traditional hardware of electrolyzers available in the market incorporates several components (PTL, current collector, and bipolar plates), increasing the hydrogen cost produced by water electrolysis. The 3D-printed reactor for lignin electrolysis eliminated the additional components and utilized the commercial macroporous nickel foam as both anode and cathode with a zero-gap configuration assembled with an anion exchange membrane (AEM). The AEM is a barrier between the anode and cathode and facilitates the ion (OH$^-$) transfer. The performance of the 3D-printed reactor for lignin electrolysis and water electrolysis was evaluated by performing electrochemical characterizations. The lignin electrolysis achieves higher current density compared to conventional water electrolysis. The effect of electrolyte flow rate on hydrogen generation sheds light on the operating parameters of the reactor. Thus, this work demonstrates a significant improvement in hydrogen generation through lignin electrolysis and shows a promising route as a new generation biomass-assisted hydrogen production method.

## 2. Materials and Method

### 2.1. Reagents

The commercial nickel foams (thickness 1.6 mm, porosity 95%, and purity 99.5%) were purchased from Goodfellow. The morphological structure of the nickel foam is given in Figure S1 (Supporting Information). Anion exchange membrane (Aemion$^{+\circledR}$) was procured from Inc, Vancouver, Canada. Lignin organosolv was kindly provided by Suzano/CÉPROCQ, Montreal (Quebec) Canada. KOH was purchased from Sigma Aldrich (Canada).

### 2.2. Cell Testing Measurement

The schematic representation for evaluating the performance of the reactor for lignin electrolysis and water electrolysis is depicted in Figure 1. The reactor for the hydrogen generation was manufactured by stereolithography (SLA)-based 3D printing. The 3D model of the reactor was designed in CAD software (Creo Parametric 8.0) and exported as an .stl file to the 3D printer (Formlabs Form 3). The model was 3D printed using the Formlabs Clear resin. The reactor obtained after 3D printing was washed thoroughly with isopropyl alcohol to remove any residual resin, followed by drying and exposure to U.V. light to complete the cross-polymerization of the resin. The 3D-printed reactor encompassed a serpentine flow channel allowing the entry and exit of the electrolyte and gas products.

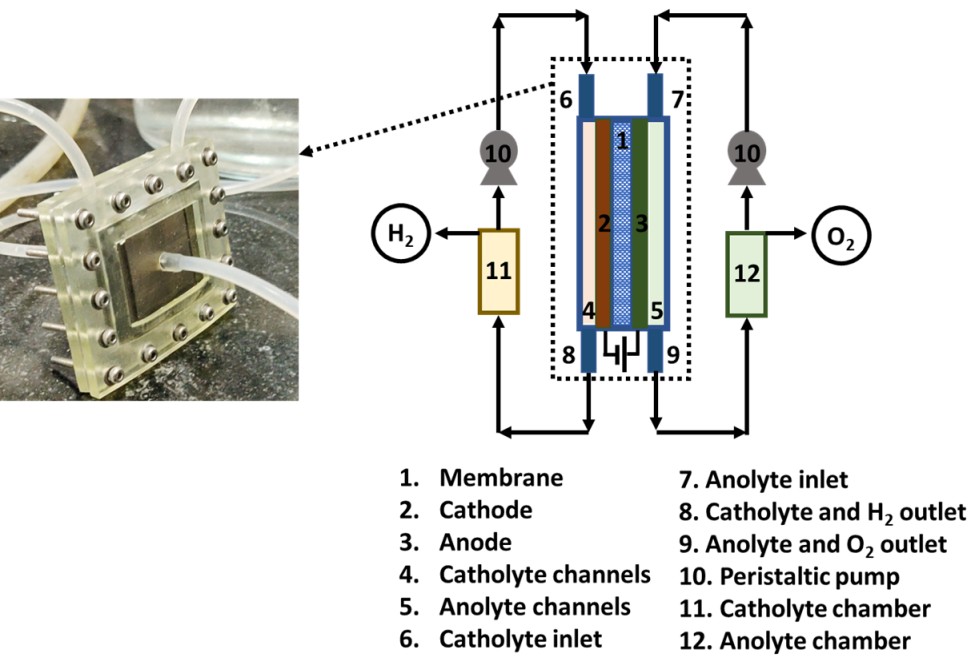

| | |
|---|---|
| 1. Membrane | 7. Anolyte inlet |
| 2. Cathode | 8. Catholyte and H$_2$ outlet |
| 3. Anode | 9. Anolyte and O$_2$ outlet |
| 4. Catholyte channels | 10. Peristaltic pump |
| 5. Anolyte channels | 11. Catholyte chamber |
| 6. Catholyte inlet | 12. Anolyte chamber |

**Figure 1.** Schematic representation of the experimental setup to evaluate the performance of lignin electrolysis and water electrolysis.

The 3D-printed reactor was assembled with the anion exchange membrane (1) sandwiched between the nickel foam as a cathode (2) and anode (3) with a 5 cm$^2$ working area. The anode and cathode were surrounded by a layer of gasket sealing the electrolyte flow. The lignin solution (10 gL$^{-1}$ in 1 M KOH) was circulated through anolyte channels (5), while cathode channels were fed with neat 1 M KOH solution (6). Under an applied voltage, the hydrogen and oxygen were generated at the cathode and anode, respectively, and transported out of the reactor through outlets 8 and 9. The electrolytic reactor was designed as a closed-loop cell system, where anolyte and catholyte flow under sealed reservoirs 11 and 12, respectively. The volume of the solution in each reservoir was 400 mL and supplied as catholyte and anolyte using a peristaltic pump (Cole Parmer Masterflex). The flow of

electrolyte solutions was varied from 4–12 mL min$^{-1}$, controlled with a peristaltic pump at room temperature.

A Biologic SP150 potentiostat/galvanostat was used to evaluate the electrochemical performance of the reactor. The reactor was conditioned by applying voltage (2 V) and waiting for the current to equilibrate, this process took 20 to 40 min. After that, the linear sweep voltammetry (LSV) was measured under various electrolyte flow rates in the voltage range of 0 to 2.5 V and the current was recorded. Potentiostatic electrochemical impedance spectroscopy (PEIS) was performed in the frequency range of 100 kHz to 10 mHz at 1.8 V and 2.2 V with an amplitude of 10 mV. The PEIS was performed at various flow rates, and the effect on the ohmic and charge transfer resistance was evaluated. Durability tests were conducted via chronoamperometry at a constant voltage of 1.8 V for 20 h for lignin electrolysis and water electrolysis, and the corresponding behavior was recorded in terms of time versus current.

## 3. Results and Discussion

Figure 1 represents the schematic device configuration of the reactor for implementing the system equipped with an anion exchange membrane (Aemion$^{+®}$) sandwiched between the nickel foam-anode and nickel foam-cathode. The electrolyzer performance was evaluated for the lignin electrolysis and water electrolysis by conducting linear sweep voltammetry (LSV) and potentiostatic electrochemical impedance spectroscopy (PEIS) at three different electrolyte flow rates of 4, 8, and 12 mL min$^{-1}$ at room temperature. Firstly, the lignin concentration was varied from 5, 10, and 15 gL$^{-1}$ to determine the concentration effect of lignin on the electrochemical activity at a 12 mL min$^{-1}$ flow rate. As shown in Figure S2 (Supporting Information), the concentration of 10 gL$^{-1}$ showed higher current, achieving 1000 mA compared to that of 5 gL$^{-1}$ (509 mA) and 15 gL$^{-1}$ (743 mA) at 2.37 V. This result indicates that the electrochemical performance of the lignin is highly dependent on the concentration of lignin. Based on the higher current, the concentration of 10 gL$^{-1}$ was selected for further electrochemical studies, the lignin electrolysis is where anolyte (10 gL$^{-1}$ in 1 M KOH) and catholyte (1 M KOH) were circulated, and showed much lower overvoltage than that of water electrolysis, feeding 1 M KOH as both anolyte and catholyte, as interpreted from the polarization curves obtained in the potential range from 0 to 2.5 V at 100 mV s$^{-1}$ (Figure 2a–c). The current obtained for the lignin electrolysis at 1.8 V with flow rates of 4, 8, and 12 mL min$^{-1}$ was 169, 197, and 202 mA, respectively, while water electrolysis with 1 M KOH showed only 88, 98, and 103 mA under similar flow rates. A similar trend was observed when the current was measured at 2.5 V for the lignin electrolysis, achieving the high output current of 684 mA (at 4 mL min$^{-1}$), 700 mA (at 8 mL min$^{-1}$), and 673 mA (at 12 mL min$^{-1}$) compared to water electrolysis, which showed low currents of 434, 530, and 569 mA under similar conditions. In addition, no mass transport limitation was observed, even at a high voltage of 2.5 V. Such high output current obtained from the lignin electrolysis is most likely due to the combination of electrooxidation of functional groups linked to the lignin such as phenolic hydroxyl and methoxy groups, the unique architecture of the reactor, controlled flow rate of electrolytes, and microporous structures of the electrodes, which promoted overall kinetics of the lignin electrolysis [19,33].

The potentiostatic electrochemical impedance spectroscopy (PEIS) was evaluated for the investigation of charge transfer resistance ($R_{ct}$) and ohmic resistance ($R_{\Omega}$) with a frequency range from 100 kHz to 10 mHz, as shown in Figure 2d–f. The $R_{ct}$ values for the lignin and water electrolysis were obtained from the corresponding semicircles of the Nyquist plots at different flow rates of 4, 8, and 12 mL min$^{-1}$. The $R_{ct}$ values calculated for the lignin electrolysis were 0.62, 0.49, and 0.41 $\Omega$, which are lower than the values obtained for water electrolysis; 1.67, 1.50, and 1.44 $\Omega$, at flow rates of 4, 8, and 12 mL min$^{-1}$, respectively. However, the $R_{\Omega}$ values for the water electrolysis under all applied flow rates were slightly lower than that of the lignin electrolysis, as seen in Figure 2d–f. This may be because of the increased viscosity of the electrolyte upon the addition of the lignin biopolymer. Based on the above PEIS analysis, it can be suggested that the presence of lignin

in an alkaline solution dramatically decreases the charge-transfer resistance, enhancing the kinetics of the lignin electrolysis for hydrogen production.

The effect of the electrolyte flow rate on the performance of the electrolyzer for water electrolysis and lignin electrolysis was also investigated by employing LSV and PEIS at 4, 8, and 12 mL min$^{-1}$ (Figure 3). The interplay between the electrolyte flow rate and the gas bubble detachment from the electrode surface determines the performance of the electrolyzer. The operating flow rate of the electrolyzer is greatly affected by the gas bubble detachment from the electrode surface, which is a balance of the interfacial tension force and the drag force acting on the gas bubble at the triple phase boundary of the electrode–electrolyte–gas bubble [34,35].

As the electrolyte flow rate was increased from 4 to 12 mL min$^{-1}$, the current at 2.5 V increased from 820 to 1027 mA for water electrolysis (Figure 3a) and 990 to 1097 mA for lignin electrolysis (Figure 3d). The increased current with an increasing flow rate of an electrolyte can be ascribed to the improvement of the mass transfer, resulting in a high rate of gas production on the electrode surface. The low electrolyte flow rate (e.g., 4 and 8 mL min$^{-1}$) at high overvoltage causes gas product accumulation on the electrode surface, resulting in a significant loss of active surface area. At a high electrolyte flow rate of 12 mL min$^{-1}$, the gas product quickly detaches from the electrode surface owing to high drag force exerted by the electrolyte. The mass transfer resistance decreases as the nucleated bubble size on the electrode surface decreases with increasing electrolyte flow rate, and the gas volume fraction in the electrolyzer compartments simultaneously reduces [35]. However, increasing the electrolyte flow rate above 12 mL min$^{-1}$ did not contribute to a significant current increase (Figure S3, Supporting Information). Therefore, the electrolyte flow rate of 12 mL min$^{-1}$ can be considered optimum for the cell operation. Increasing the electrolyte flow rate above 12 mL min$^{-1}$ will induce an additional power consumption associated with pumping of the electrolyte, which will have a detrimental effect on the cost of hydrogen produced.

The change in the current with the electrolyte flow rate was related to the $R_\Omega$ and $R_{ct}$ by obtaining PEIS at 1.8 V and 2.2 V at different electrolyte flow rates. The current and the electrolyte flow rate regulates the two-phase flow behavior and the gas volume fraction in the electrolyzer. The electrolyte and the electrical connections contribute to the $R_\Omega$. Moreover, in an electrolyzer, a two-phase flow induced under an applied voltage generates gas bubbles on the electrode surface, impeding the electron transfer. The $R_\Omega$ for the water electrolysis remained almost unchanged (0.27–0.29 $\Omega$ at 1.8 V and 0.8–0.31 $\Omega$ at 2.2 V) for all the electrolyte flow rates (Figure 3b,c). Similarly, the $R_\Omega$ was 0.38 $\Omega$ and 0.34 $\Omega$ at 1.8 V and 2.2 V, respectively, and remained unchanged at all electrolyte flow rates for the lignin electrolysis. A high electrolyte flow rate increases the interfacial area of the gas bubbles, which results in rapid detachment from the electrode surface without coalescing [36]. This results in low $R_\Omega$ because the bubble gets detached at a small size allowing the electrolyte to come in contact with the electrode. The $R_{ct}$ for water electrolysis decreased from 2.62 to 1.91 $\Omega$ at 1.8 V, and that at 2.2 V decreased from 0.76–0.54 $\Omega$ (Figure 3b,c). For lignin electrolysis, the $R_{ct}$ decreased from 1.03 to 0.81 $\Omega$ at 1.8 V, and that at 2.2 V decreased from 0.54 to 0.46 $\Omega$ as the electrolyte flow increased from 4 to 12 mL min$^{-1}$ (Figure 3e,f). The change in the $R_{ct}$ is more significant with the flow rate at 2.2 V compared to that at 1.8 V for both water electrolysis and lignin electrolysis. This is owing to the low perturbation in the electrolyte flow as fewer gas products are generated at a low current corresponding to 1.8 V. At high electrolyzer operating voltages, the gas bubbles generate vigorously, reducing the ion transfer to the reaction sites. The high electrolyte flow rate enables rapid removal of the generated gas products, which is imperative to lower the $R_{ct}$.

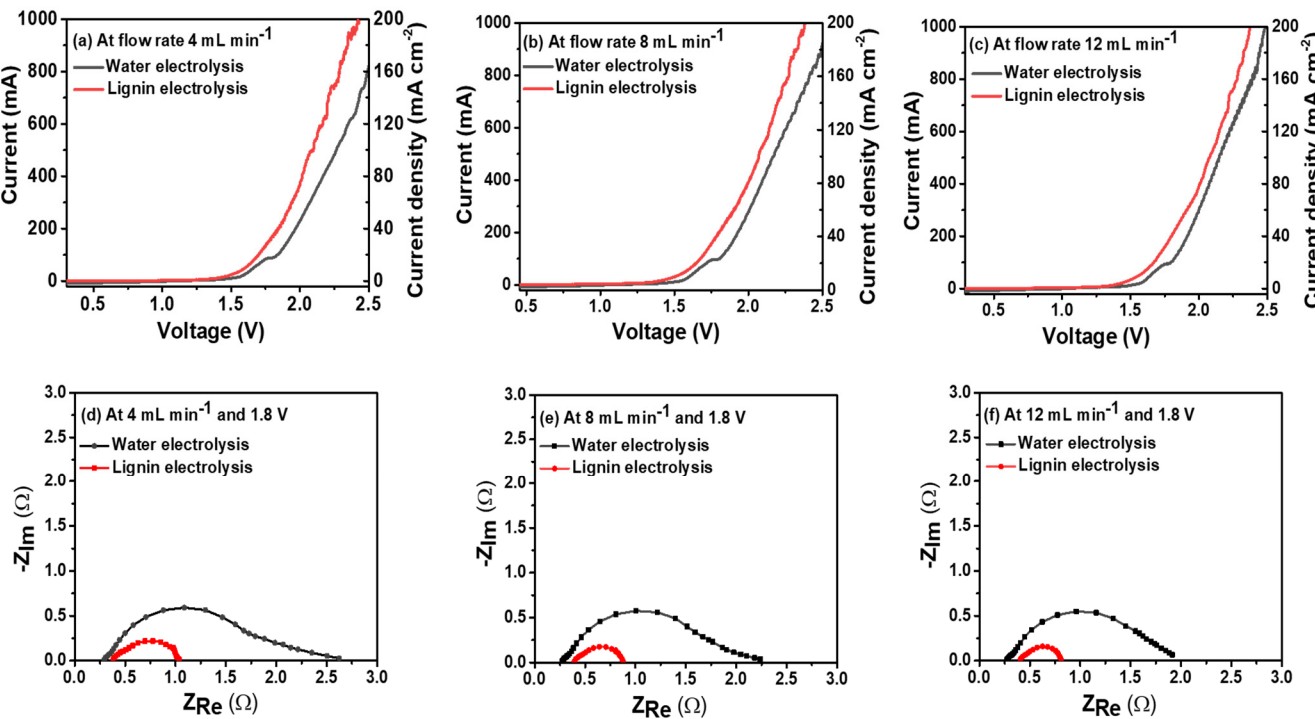

**Figure 2.** Linear sweep voltammograms (LSVs) comparing the difference in overvoltage and current density between water electrolysis and lignin electrolysis at (**a**) 4 mL min$^{-1}$, (**b**) 8 mL min$^{-1}$, and (**c**) 12 mL min$^{-1}$. The electrochemical impedance spectroscopy (PEIS) at 1.8 V comparing the ohmic and charge transfer resistance between the water electrolysis and lignin electrolysis at (**d**) 4 mL min$^{-1}$, (**e**) 8 mL min$^{-1}$, and (**f**) 12 mL min$^{-1}$.

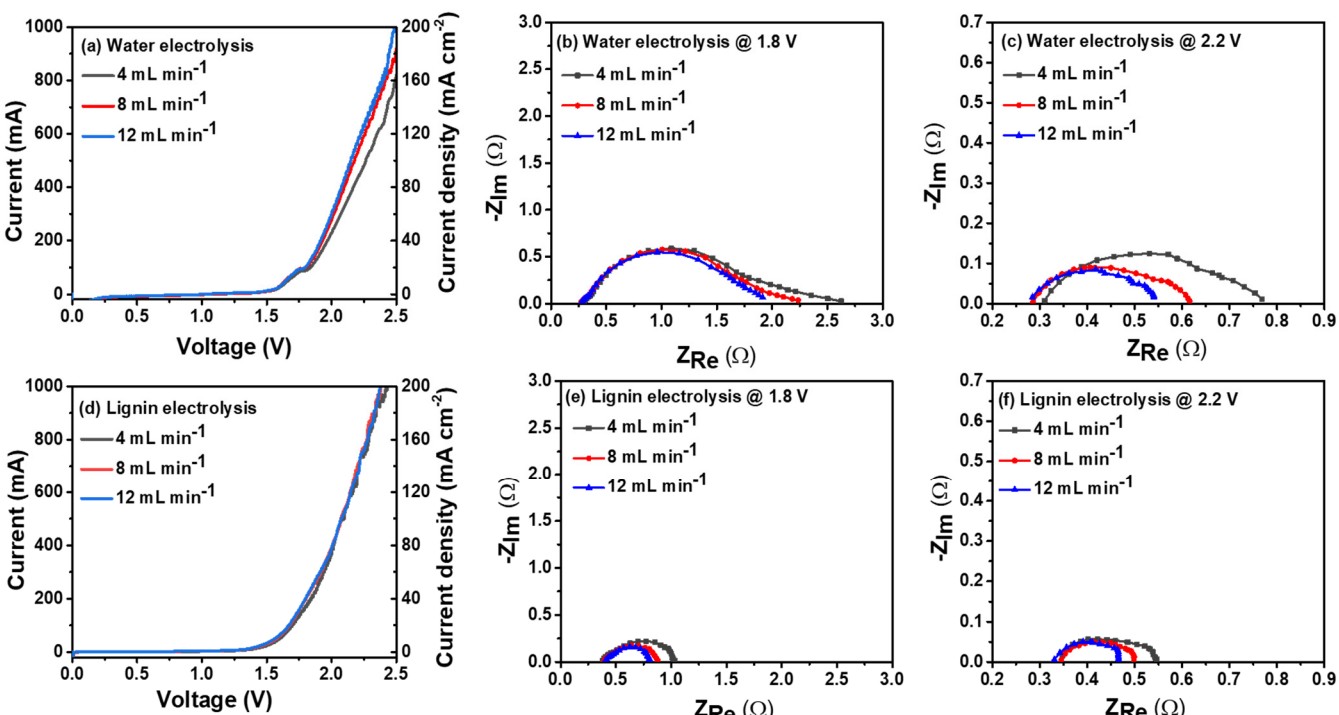

**Figure 3.** Effect of flow rate on the performance determined by performing linear sweep voltammogram (LSV) for (**a**) water electrolysis and (**d**) lignin electrolysis. The change in the ohmic and charge transfer resistance concerning the flow rate is determined by electrochemical impedance spectroscopy (PEIS) for water electrolysis (**b,c**) and lignin electrolysis (**e,f**).

The current profile concerning time was obtained for the water electrolysis and lignin electrolysis by performing chronoamperometry at a constant applied voltage of 1.8 V at 12 mL min$^{-1}$ electrolyte flow rate (Figure 4). The water electrolysis attained a constant current during the chronoamperometric operation of 20 h. The current obtained from the lignin electrolysis showed a high current at the start of the test. A higher current is directly indicative of the high rate of lignin electrooxidation. However, lignin electrolysis experienced an attenuation in current throughout chronoamperometric operation under a constant applied voltage of 1.8 V. The anode in the lignin electrolysis was operated in batch mode with the recycling of the anolyte. This led to the consumption of the reacting species participating in the lignin electrooxidation, which was higher in the beginning, and later on, their concentration became low near the electrode surface [16,17]. After the complete conversion of the lignin in the anolyte due to recycling, the OER switches from lignin oxidation to water oxidation, characterized by a reduction in the kinetics of water splitting for hydrogen generation.

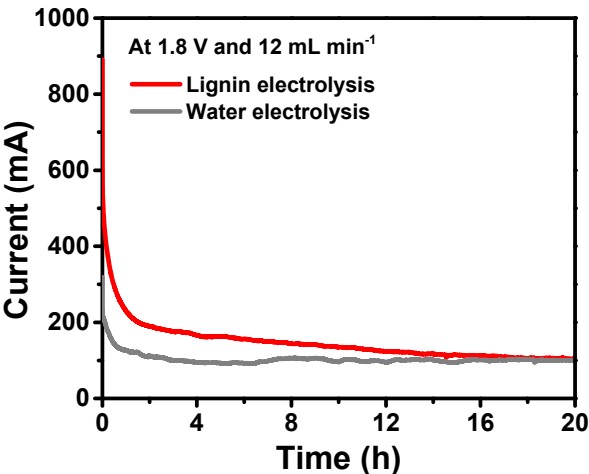

**Figure 4.** Chronoamperometry at a constant applied voltage of 1.8 V and 12 mL min$^{-1}$ electrolyte flow rate for water electrolysis and lignin electrolysis.

Furthermore, the area under the curve of the chronoamperometry graph in Figure 4 is calculated to estimate the charge in coulombs. The charge associated with lignin electrolysis is 10,604 C, which is approximately 1.38 times greater than that obtained for the water electrolysis (7656 C). The corresponding rate of hydrogen production was evaluated from Faraday's law. The hydrogen production rate was 1.38 mL min$^{-1}$ for lignin electrolysis which is high in contrast to that of water electrolysis (0.8 mL min$^{-1}$). Considering recent and future development of biomass reforming for hydrogen production, the advantages of lignin electrolysis are clear, it would facilitate the storage of energy in the form of hydrogen through electrolysis at lower overvoltage than traditional water electrolysis.

## 4. Conclusions

A sustainable hydrogen production route by lignin-assisted electrolysis was demonstrated at room temperature in a 3D-printed flow reactor. Lignin electrolysis reduced the overvoltage of water electrolysis compared to conventional water oxidation at the anode. The electrolyte flow rate highly influenced the performance of the reactor. The current at 2.5 V for the lignin electrolysis increased from 990 mA to 1097 mA with an increase in flow rate from 4 to 12 mL min$^{-1}$. The increase in the current density concerning the electrolyte flow rate was correlated with the decrease in the charge transfer resistance. The mass transfer resistance in the reactor arising from the two-phase flow of the electrolyte and gas bubbles was alleviated at a flow rate of 12 mL min$^{-1}$. The hydrogen production rate was 1.1 mL min$^{-1}$ and 0.8 mL min$^{-1}$ obtained by lignin electrolysis and water electrolysis, respectively. The flow reactor for the lignin electrolysis enables continuous biomass

conversion alongside hydrogen generation and paves the way for future research on the scale-up strategy.

**Supplementary Materials:** The following supporting information can be downloaded at: https://www.mdpi.com/article/10.3390/catal12121646/s1, Figure S1: Scanning electron microscope images of nickel foam at different magnifications; Figure S2: LSV curves for lignin electrolysis with different concentrations of lignin at scan rate of 100 mV s-1; Figure S3: Effect on electrolyte flow rate on water electrolysis performance.

**Author Contributions:** Conceptualization, M.K. and B.S.D.; methodology, M.K. and B.S.D.; draft preparation, M.K. and B.S.D.; writing-reviewing and editing, M.K. and B.S.D.; formal analysis, M.K. and B.S.D.; visualization, A.S.; visualization, S.S.; supervision, S.S.; project administration, S.S.; funding acquisition, S.S. All authors have read and agreed to the published version of the manuscript.

**Funding:** This research was funded under Canada Research Chairs Program.

**Data Availability Statement:** The data used to support the findings of this research work are included within the article and supporting information, more data may be provided on reasonable request from corresponding author.

**Conflicts of Interest:** The authors declared no conflict of interests.

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
