# Peer review of "Lignin Electrolysis at Room Temperature on Nickel Foam for Hydrogen Generation: Performance Evaluation and Effect of Flow Rate"

_catalysts, doi:10.3390/catal12121646_

Round 1

Reviewer 1 Report

In the manuscript “Lignin electrolysis at room temperature on nickel foam for hydrogen generation: Performance evaluation and effect of flow rate”, In this manuscript, the author reported the research on hydrogen production from lignin assisted electrolysis of water. This is not a novel preparation technology, and the results is not very prominent. The data analysis is insufficient and the discussion is not clear. I suggest he article should be transferred to the other journal.

Author Response

Reviewer 1# In the manuscript “Lignin electrolysis at room temperature on nickel foam for hydrogen generation: Performance evaluation and effect of flow rate”, In this manuscript, the author reported the research on hydrogen production from lignin-assisted electrolysis of water. This is not a novel preparation technology, and the results is not very prominent. The data analysis is insufficient and the discussion is not clear. I suggest he article should be transferred to the other journal.

Response: We thank the reviewer for the comment on our manuscript. Despite the unfavorable remark of the reviewer, we strongly believe that the approach we have proposed in this work for lignin electrolysis is technologically new, which does not rely on expensive additional key components such as porous transport layer, current collector, and bipolar plates like conventional water electrolysis by heavy weight hardware. In addition, using robust porous nickel foam as both anode and cathode eliminate the requirements of binder and/or ionomer, allowing good mechanical stability and high utilization of active phase of the electrodes. The present research paves the way for scaleup of the device for the continuous hydrogen production by lignin electrolysis.        

Reviewer 2 Report

In the present manuscript the authors reported a simple yet effective approach for hydrogen generation by lignin electrolysis at room temperature, utilizing low cost nickel foam as electrodes and AEM in a 3D printed reactor. The performance of the electrolyzer involving lignin electrolysis was evaluated by means of LSV and PEIS at different flow rate and was compared with that involves water electrolysis. The data obtained are competently analyzed and interpreted. The topic of the manuscript is adequate for Catalysis. The article is generally well written and I would suggest to be accepted as it is.

Author Response

Reviewer 2# In the present manuscript the authors reported a simple yet effective approach for hydrogen generation by lignin electrolysis at room temperature, utilizing low-cost nickel foam as electrodes and AEM in a 3D printed reactor. The performance of the electrolyzer involving lignin electrolysis was evaluated by means of LSV and PEIS at different flow rate and was compared with that involves water electrolysis. The data obtained are competently analyzed and interpreted. The topic of the manuscript is adequate for Catalysis. The article is generally well written, and I would suggest to be accepted as it is.

Response: We thank the reviewer for the positive remark on our manuscript.

Reviewer 3 Report

The authors studied the lignin electrolysis on Ni foam for hydrogen generation. However, more results and discussion should be provided for maintaining the integrality of this manuscript. Therefore, after carefully evaluate this work, the reviewer suggest that this paper can be accepted after addressing the following issues.

1.       The 3D printed reactor with detailed information should be shown by a high-resolution picture by camera or other devices.

2.       The lignin solution with different concentrations (such as 5 gL-1 and 15 gL-1) should be given and discussed through the manuscript.

3.       More detailed information of Ni foam should be given, such as the morphology (SEM images), length, width, purity, and porosity.

4.       The commercial RuO2 or other OER catalysts should be used for the comparison purpose.

5.       The electrochemical performance of the water electrolysis with the flow rate of 16 mL min-1 should be considered for finding the optimal conditions in this manuscript.

6.       The chronoamperometry test should be extended for at least 20 h.

Round 2

Reviewer 1 Report

The new version has sorted out most problem. I think it can be published.

Reviewer 3 Report

The authors have addressed all my comments and I suggest its publication without further revision.